# Targeting Hypoglycemic Natural Products from the Cloud Forest Plants Using Chemotaxonomic Computer-Assisted Selection

**DOI:** 10.3390/ijms252010881

**Published:** 2024-10-10

**Authors:** Cecilia I. Mayo-Montor, Abraham Vidal-Limon, Víctor Manuel Loyola-Vargas, Oscar Carmona-Hernández, José Martín Barreda-Castillo, Juan L. Monribot-Villanueva, José A. Guerrero-Analco

**Affiliations:** 1Red de Estudios Moleculares Avanzados, Instituto de Ecología A.C., Xalapa 91073, Mexico; cecilia.mayo@posgrado.ecologia.edu.mx (C.I.M.-M.); abraham.vidal@inecol.mx (A.V.-L.); jose.barreda@posgrado.ecologia.edu.mx (J.M.B.-C.); 2Unidad de Biología Integrativa, Centro de Investigación Científica de Yucatán, Mérida 97205, Mexico; vmloyola@cicy.mx; 3Facultad de Biología, Universidad Veracruzana, Xalapa 91090, Mexico; ocarmona@uv.mx

**Keywords:** ligand-based virtual screening, dipeptidyl peptidase IV, α-amylase, α-glucosidase, plant metabolomics, ensemble molecular docking

## Abstract

The cloud forest (CF), a hugely biodiverse ecosystem, is a hotspot of unexplored plants with potential for discovering pharmacologically active compounds. Without sufficient ethnopharmacological information, developing strategies for rationally selecting plants for experimental studies is crucial. With this goal, a CF metabolites library was created, and a ligand-based virtual screening was conducted to identify molecules with potential hypoglycemic activity. From the most promising botanical families, plants were collected, methanolic extracts were prepared, and hypoglycemic activity was evaluated through in vitro enzyme inhibition assays on α-amylase, α-glucosidase, and dipeptidyl peptidase IV (DPP-IV). Metabolomic analyses were performed to identify the dominant metabolites in the species with the best inhibitory activity profile, and their affinity for the molecular targets was evaluated using ensemble molecular docking. This strategy led to the identification of twelve plants (in four botanical families) with hypoglycemic activity. *Sida rhombifolia* (Malvaceae) stood out for its DPP-IV selective inhibition *versus S. glabra*. A comparison of chemical profiles led to the annotation of twenty-seven metabolites over-accumulated in *S. rhombifolia* compared to *S. glabra*, among which acanthoside D and *cis*-tiliroside were noteworthy for their potential selective inhibition due to their specific intermolecular interactions with relevant amino acids of DPP-IV. The workflow used in this study presents a novel targeting strategy for identifying novel bioactive natural sources, which can complement the conventional selection criteria used in Natural Product Chemistry.

## 1. Introduction

Diabetes mellitus (DM) is a metabolic disorder of heterogeneous etiology characterized by chronic hyperglycemia, which in the long term produces renal failure, blindness, slow-healing wounds, and arterial diseases [1,2]. DM is considered one of the fastest-growing health emergencies of the 21st century: in 2021, 537 million DM patients were recorded, and it is projected to increase to 643 million by 2030 and 783 million by 2045 [3]. Type II DM (DMII) is caused by alterations in the secretion and action of insulin and is the most frequent type, with 90–95% of cases. Due to chronic hyperglycemia, the pharmacotherapy used for its treatment consists mainly of administering hypoglycemic oral agents that act through different mechanisms of action [4]. Inhibition of α-amylase (αA) and α-glucosidase (αG) enzymes delays the digestion of complex carbohydrates such as starch and prevents sudden increases (spikes) in blood glucose levels [4,5,6]. Acarbose and voglibose are inhibitors of both enzymes. However, they cause adverse side effects such as flatulence, diarrhea, and abdominal pain [5,7]. In addition to these targets, the inhibition of the dipeptidyl peptidase IV enzyme (DPP-IV) prevents the hydrolysis of incretin hormones, glucagon-like peptide-1 (GLP-1), and gastric inhibitory polypeptide (GIP) [8]. Among the functions of these hormones is the activation of separate G protein-coupled receptors on pancreatic β cells, associated with insulin secretion through exocytosis. GLP-1 also reduces glucagon secretion, delays gastric emptying, and produces a satiety effect [9]. DPP-IV inhibition produces a secondary anti-hyperglycemic effect caused by the action of endogenous substrates. Gliptins (sitagliptin, vildagliptin, saxagliptin, alogliptin, and linagliptin) are a group of DPP-IV inhibitors that do not present intrinsic hypoglycemic risk. However, they cause other adverse side effects, such as nasopharyngitis, headache, nausea, and skin sensitivity [10,11].

Due to its undesirable side effect profiles and eventual decrease in effectiveness, it is currently necessary to conduct scientific studies to search for alternatives in which better therapeutic management of DMII can be carried out [12]. In this regard, it is well known that plants and their derivatives (extracts, fractions, and pure compounds) are promising and well-accepted alternatives for controlling DMII in different cultures worldwide [13]. Plant natural products are characterized by presenting a wide structural diversity optimized to fulfill biological functions, such as regulating mechanisms involved in biotic interactions [14]. Due to structural domain conservation in macromolecular targets across many organisms, some plant natural products are recognized by human protein targets with therapeutic relevance [15,16]. Conventionally, in Natural Products Chemistry (NPC), the selection of plant materials for bioactive compounds discovery involves criteria such as ethnopharmacological (uses in traditional medicine) [17], random (plant material availability) [18], chemotaxonomic (plants phylogenetically related to other bioactive plants) [19], and, recently, the application of chemical–computational tools (in silico prediction of bioactivity) [20].

Chemotaxonomic classification is an approach used to organize taxa based on related biosynthetic pathways and chemical structures of specialized metabolites, which is restricted to phylogenetically related organisms [21]. In addition, regarding the techniques used in computational criteria, virtual screening consists of reducing a library of compounds to a limited number of potentially bioactive molecules [22]. Since its first report in 1997, the applications of virtual screening have been increasing thanks to the increase in data processing capacity, the structural characterization of macromolecules, and the availability of libraries of chemically diverse compounds [23,24]. Currently, using computational tools in the study of the pharmacological properties of natural products has increased the success rates in bioactive molecule identification in academia as well as at the industrial level. Furthermore, it has reduced the investment of resources related to the introduction of new drugs to the market [25]. Nevertheless, these computational approaches continue to require the integration of robust experimental methods to validate their predictive capacity, which helps refine the virtual models to produce data characterized by predictability, computability, and stability [26].

In Mexico, cloud forest (CF) is considered the ecosystem with the highest biodiversity per surface area unit, and it covers approximately 1% of the total territory of Mexico, comprising 10% of the plant species described for the country (*circa* 30% endemic) [27]. CF has also been identified as an unexplored niche of plants with phytomedicinal potential [28]. In recent years, our workgroup has described the antibacterial, antifungal, and insecticidal potential of randomly selected CF plants in Veracruz, Mexico [18,29]. In these studies, success rates in identifying those with promising activities yielded rates of from 13.3 to 76.5% of the total studied plants, and these species can now be used in strategies for developing new control agents against pathogens that affect crops of commercial interest [30]. However, there is still a gap in ethnobotanical knowledge about CF species, which has limited their application in studies focused on determining their potential therapeutic activities.

This work aimed to establish a combined chemotaxonomic and computational criterion to rationally target CF plants with potential hypoglycemic activity. The application of this approach allowed the identification of plant species with inhibitory activity in DPP-IV, αA, and αG from a highly biodiverse ecosystem, indicating that it could be used for bioprospecting purposes in a complementary way to conventional selection approaches in NPC.

## 2. Results

### 2.1. Rational Selection of the Cloud Forest Botanical Families with Potential Hypoglycemic Activity

Using the Plant Metabolic Network (PMN) library for plants to identify the reported families for the CF, we retrieved 13 families, including 43 species and 97,094 metabolites. After the pre-processing step (see Section 4), the library was enriched to 129,119 metabolites (100%) with defined stereochemistry. The library consisted of primary metabolites (peptides, nitrogenated bases, carbohydrates, and lipids) and specialized metabolites (amine-containing metabolites, phenolic compounds, and terpenoids). In the ligand-based virtual screening (LBVS), a total of 548 metabolites (0.4% from the prepared library) distributed in the three natural products groups (amine-containing metabolites, phenolics, and terpenoids) accomplished all criteria used for filtering (see Section 4). Thus, this final set of compounds was selected due to its potential hypoglycemic activity (Figure 1). The search for these molecules in the initial library allowed us to rank the 13 botanical families based on the percentage of the potential hypoglycemic metabolites presented by the group. For each case, the two families with the highest percentage were selected (Table 1). Considering all the data, the Brassicaceae, Fagaceae, Euphorbiaceae, Solanaceae, and Malvaceae families were the best candidates for the experimental evaluation of hypoglycemic activity.

### 2.2. Enzymatic Inhibition Activity

Twelve species belonging to the Euphorbiaceae, Fagaceae, Malvaceae, and Solanaceae families were recollected in the two study sites (Brassicaceae species were not found). Specimens of all species were deposited in the collection of Herbarium XAL (see Section 4.3). The inhibitory enzymatic activity of their crude extracts was assessed using in vitro assays (Figure 2).

The percentage of αA inhibition ranged from 1.8 ± 1.49% (*Sida rhombifolia*) to 98.6 ± 0.98% (*Acalypha alopecuroidea*), and nine species (75%) displayed an inhibition higher than 50% (Figure 2A). Also, for this enzyme, *A. alopecuroidea* (Euphorbiaceae), *Quercus lancifolia,* and *Q. xalapensis* (Fagaceae) presented the same inhibitory activity as the positive control (3 mM acarbose≈ 100% inhibition). According to the Dunnett test, they did not present statistically significant differences (Figure 2A). The αG inhibition percentages were observed from 6.9 ± 0.59% (*S. rhombifolia*) to 99.9 ± 0.31 (*Pavonia schiedeana*), and eight species (67%) exhibited more than 50% inhibition, which is in concordance with those species identified with this criterion for αA [only exception: *Solanum lanceolatum* (Solanaceae)] (Figure 2B). In addition, six species also showed a higher percentage of inhibition compared to the positive control (30 mM acarbose= 80.1 ± 0.19%), among which *Q. lancifolia*, *P. schiedeana,* and *Malvaviscus arboreous* (Malvaceae) stand out due to the fact that they reached an inhibition of around 100% (Figure 2B). Finally, the DPP-IV enzyme’s inhibition percentages ranged between 16.4 ± 8.71 (*P. schiedeana*) and 56.3 ± 3.37% (*Q. xalapensis*). Three species (25%) presented more than 50% inhibition: *Q. xalapensis*, *S. myriacanthum*, and *S. lanceolatum* (Figure 2C). From the total tested extracts, only those from *Sida* spp. (Malvaceae) presented a selective inhibition profile when considering the three enzymes. *S. rhombifolia* reached an inhibition of 39.2 ± 2.41% in DPP-IV, compared to 1.8 ± 1.49% and 6.9 ± 0.60% for αA and αG, respectively. On the other hand, *S. glabra* was consistent for DPP-IV (47.4 ± 2.07%), but presented higher inhibition on αA and αG (50.0 ± 1.29% and 62.5 ± 1.32%, respectively).

### 2.3. Determination of Half Maximal Inhibitory Concentration and Half Maximal Lethal Concentration of Selected Extracts

The selective inhibition profile towards DPP-IV of *S. rhombifolia* concerning *S. glabra* was confirmed by determining the half-maximal inhibitory concentration (IC_50_) of their extracts (Table 2). Consistently with the preliminary evaluation, *S. rhombifolia* and *S. glabra* exhibited similar inhibitory activity in DPP-IV (IC_50_ = 2.67 and 2.26 mg/mL, respectively). On the other hand, *S. rhombifolia* showed an IC_50_ higher than 4 mg/mL in αA and αG, while *S. glabra* showed lower values than 2 mg/mL for both enzymes. It is worth mentioning that *S. glabra* showed higher inhibitory activity on αG compared to the positive control (0.33 *vs*. 2.88 mg/mL). In terms of toxicity, *S. rhombifolia* (1.41 mg/mL) and *S. glabra* (>2 mg/mL) displayed higher half-maximal lethal concentration (LC_50_) values than the positive control colchicine (0.55 mg/mL). Both extracts were classified as non-toxic according to the Clarkson index (LC_50_ > 1.0 mg/mL non-toxic, 0.5 < LC_50_ ≤ 1.0 mg/mL low toxicity, 0.1 < LC_50_ ≤ 0.5 mg/mL medium toxicity and LC_50_ ≤ 0.1 mg/mL high toxicity) [31].

Given the resulting biological activities and the intrinsic phylogenetic relationship between *S. rhombifolia* and *S. glabra*, these results raised the possibility of identifying metabolites accumulated in *S. rhombifolia* compared to *S. glabra*, with potential selective inhibition for DPP-IV.

### 2.4. Comparative Untargeted Metabolomic Analysis of Active Species

The untargeted metabolomic analysis of the extracts of *S. rhombifolia* and *S. glabra* allowed us to obtain a combined spectra database of both ionization modes (ESI^+^ and ESI^−^) with 2109 rt-*m*/*z* values (spectrometric features). Two main signal groups were observed in the heatmap built with this database (Figure 3A), suggesting different chemical profiles between these species. The species-based distribution of rt-*m*/*z* values consisted of 1172 unique spectrometric features for *S. glabra*, 275 unique features for *S. rhombifolia*, and 662 shared features, indicating that *S. rhombifolia* shares about 70% of its global metabolic profile with *S. glabra* (Figure 3B). The pathway analysis revealed the tentative presence of precursors and products for the biosynthesis of flavonoids, flavones, flavonols, monoterpenoids, diterpenoids, phenylpropanoids, and isoquinoline alkaloids (Figure 3C and Appendix A). Through the fold change analysis, 319 rt-*m*/*z* features were found that could correspond to metabolites or their fragments over-accumulated in *S. rhombifolia* compared to *S. glabra* (fold change ≥ 2, *p* ≤ 0.05), which could be responsible for its selective inhibitory activity towards DPP-IV (Figure 3D). The search for experimental exact mass values in different public spectral databases allowed us to tentatively identify twenty-two over-accumulated specialized metabolites that have been described in plants belonging to the Malvaceae family (59%), and other species that coincide with *S. rhombifolia* at the malvids clade level (41%) (Table 3). These compounds were identified with confidence level two (putative annotation) compared with MS/MS spectral data (Appendix A).

### 2.5. Identification and Quantification of Phenolic Compounds

Using the untargeted metabolomic approach, different phenolic compounds were tentatively identified, followed by a phenolics-targeted metabolomic analysis. One phenolic precursor and 29 phenolic compounds belonging to phenolic acids, phenylpropanoids, flavonoids, coumarins, and lignans were identified and quantified. Of them, 24 were identified in *S. rhombifolia* and 29 in *S. glabra*. In addition, the fold change analysis allowed the identification of five metabolites over-accumulated in *S. rhombifolia* compared to *S. glabra* (fold change ≥ 2 and *p* < 0.05). These were kuromanin, naringin, vanillin, vanillic acid, and *trans*-cinnamic acid (Table 4).

### 2.6. Ensemble Molecular Docking

The different metabolomic analyses allowed us to identify 27 specialized metabolites (untargeted approach: 22, phenolics-targeted approach: five) over-accumulated in *S. rhombifolia* compared to *S. glabra*. Through ensemble molecular docking, the affinity of the identified metabolites for the enzymes DPP-IV, αA, and αG was evaluated based on the mean binding free energy (BFE) and compared to commercial drugs used as controls (Appendix A). For the three enzymes, the negative control metformin presented the lowest affinity, obtaining the highest mean BFE value (DPP-IV = −3.5 ± 0.17, αA = −3.5 ± 0.30, and αG = −4.0 ± 0.27 kcal/mol). Meanwhile, among the positive controls, sitagliptin (reversible competitive inhibitor) presented a mean BFE value of −8.0 ± 0.25 kcal/mol on DPP-IV. On the other hand, acarbose (reversible competitive inhibitor) presented BFE values of −8.1 ± 0.18 and −8.5 ± 0.41 kcal/mol for αA and αG, respectively. Notably, below the BFE values for positive controls, 13 molecules were observed with potentially higher affinity for the active site of DPP-IV. In contrast, 12 and nine molecules exhibited BFE values below positive controls for αA for αG, respectively (Figure 4).

To further investigate the binding modes between the identified metabolites and the three molecular targets, ten compounds with the lowest BFE values were selected. These were *trans*-tiliroside, *cis*-tiliroside, depressonol A, naringin, kaempferitrin, acanthoside D, thermopsoside, kuromanin, corchorusoside D, and rhaponticin. Finally, by calculating the intermolecular interactions (Appendix A), it was observed that nine of the selected metabolites established hydrogen bonds and Pi–Pi stacking interactions with amino acids of the S1 (His 708), S2 (Arg 93), S2 extensive (Phe 325, Arg 326, Trp 595) subsites, and, to a lesser extent, with the catalytic triad of DPP-IV (His 708) [51]. In contrast, only four metabolites exhibited interactions with the catalytic triads of αA (Asp 196, Glu 232, Asp 299) [52] and αG (Asp 212, Glu 274, Asp 349) [53] (Table 5). Among these metabolites, acanthoside D and *cis*-tiliroside were particularly interesting since they established hydrogen bonds and Pi–Pi stacking interactions with His 708 and Trp 595 of DPP-IV. On the other hand, they did not show intermolecular interactions with any of the amino acids of the catalytic triad of αA and αG (Figure 5).

## 3. Discussion

Within our research group, bioprospecting studies have recently been carried out in the CF of Veracruz (Mexico). Through the random selection criterion, species with antibacterial, antifungal, and insecticidal activity have been identified with the potential to be used as agents for biological control of phytopathogens and vectors, with success percentages ranging from 13.3 to 76.5% [18,29]. In this work, a bioprospecting study focused on the therapeutic applications of plants was carried out for the first time in this ecosystem. When pharmacological assays are used in combination with random selection criteria in regions with high biodiversity and endemism [54], lower success percentages have been obtained compared to the ethnopharmacological selection criterion [55,56]. For example, in the search for plants in the tropical rainforest of the Amazon with antimycobacterial activity, 50% of the species with uses in traditional medicine were active compared to 16.7% of those randomly selected [57]. The diversity of species that allowed the chemotaxonomic computer-assisted approach to be selected in this work and the high inhibitory activity exhibited by their extracts provide evidence of its successful predictive capacity. In that sense, this strategy is placed as a complementary alternative to the conventional criteria that could be used to discover plants with therapeutic applications. When considering a cut-off criterion of 50% of enzymatic inhibition, success percentages of 25% (DPP-IV), 67% (αG), and 75% (αA) were obtained, which exceed the results previously reported with the random approach. Furthermore, the DPP-IV inhibitory activity of these 12 CF species is reported for the first time, making them good candidates for bioassay- or metabolomic-guided isolation to obtain potential new natural agents for controlling DMII.

For αA and αG inhibition results, the extract of the aerial parts of *R. communis* in the Euphorbiaceae family presented an inhibitory activity of αA (75.5 ± 1.31%) similar to that previously described for the *n*-butanol extract from the roots (68%) [58]. In this work, the inhibitory activity of *R. communis* in αG is reported for the first time (96.4 ± 0.66%) and placed in the group of the best inhibitors of this enzyme. It must be highlighted that *R. communis* is used in treating diabetes in Mexico [59]; hence, its inhibitory profile supports its use in traditional medicine and suggests that its components could act through the inhibition of αA and αG.

For the Solanaceae family, similar results were obtained for *S. betaceum* to those described in the literature, in which the aqueous extract of the leaves was not active in αA and was moderately active in αG (IC_50_ = 1.617 mg/mL) [60]. This pattern coincides with the observed in this work (αA = 4.5 ± 3.14%, αG = 25.6 ± 2.84%, DPP-IV = 31.9 ± 1.86%). On the other hand, the extract of the aerial parts of *S. nigrum* inhibited αA by 46.8 ± 1.40%, compared to 81.76 ± 1.4% reported for the methanolic extract of the whole plant [61]. Specialized metabolites regulate the chemical interaction between plants and their environment. Additionally, their biosynthesis is temporal and tissue-differential as an adaptive mechanism, allowing plants to respond to the chemical information transference with other organisms [62,63,64]. Within the genus *Solanum, S. incanum, S. surattense*, and *S. torvum* roots are traditionally used in DM treatment, and their hypoglycemic activity has been evaluated through in vivo experiments [65,66,67]. There is literature evidence about the inhibitory capacity of αA by *S. nigrum*, and the hypoglycemic properties described for the roots of *Solanum* species suggest the presence of metabolites accumulated in *S. nigrum* tissues, which were not evaluated in this work, which could present inhibitory activity in the enzymes of interest.

Within the Malvaceae family, *M. arboreus* exhibited the best inhibitory profile when considering αA (90.6 ± 0.23%) and αG (99.4 ± 0.84%). In contrast, in a previous study, the aqueous extract of *M. arboreus* leaves from Yucatan (Mexico) inhibited approximately 20% of the activity of αA and αG [68]. This lower effect could be attributed to the solvent used in the metabolite extraction or the environmental influence on metabolite biosynthesis. Individuals of the same species subject to different abiotic conditions may present differences in specialized metabolite profiles [64]. The Mexican CF has an archipelago-type geographic distribution, where constant atmospheric saturation by water vapor produces its characteristic high relative humidity, manifested as clouds and fog. Besides its biogeographic history, fragmented distribution and close contact with other vegetation types are proposed as factors involved in the development of its high biodiversity [69]. The combination of these biotic and abiotic factors could influence the production of specialized metabolites to contribute to the adaptability of the CF plants and modify their biological activity profiles compared with specimens from other ecosystems [29].

Among the phylogenetically related species evaluated in this work, only *S. rhombifolia* showed a selective inhibition profile towards DPP-IV compared to *S. glabra*. This effect is desirable because, due to its mechanism of action, therapy with DPP-IV inhibitors does not present an intrinsic risk of hypoglycemia and has a favorable side effect profile compared to those of αA and αG inhibition [8]. In addition, the inhibitory activity presented by *S. rhombifolia* and *S. glabra* was interesting since the systematic analysis of ethnobotanical patterns has recently shown that taxonomically related plants cover a similar chemical space, which correlates with their therapeutic applications. Based on this chemotaxonomic premise, plants with anti-hyperglycemic activity have been identified [70,71].

As far as we know, *S. rhombifolia* is not used in traditional Mexican medicine for treating diabetes. However, it is used for this purpose in both Brazil and India [72]. In the pharmacological context, *S. rhombifolia* is a stand-out plant due to its antioxidant [43,73], anti-inflammatory [43,74], and antifungal properties [75]. For hypoglycemic activity, the ethanolic extract of the aerial parts presented inhibitory activity in vitro in both αA (IC_50_ = 831.76 µg/mL) and αG (IC_50_ = 1202.3 µg/mL). Qualitative phytochemical analysis showed that this extract contained phenols, flavonoids, glycosides, tannins, saponins, and steroids [76]. On the other hand, the acetone extract of the aerial parts inhibited the activity of αG from *Saccharomyces cerevisiae* (IC_50_ = 8.1 ± 0.34 µg/mL). From this extract, a β-sitosterol and stigmasterol mixture, 20-hydroxyecdysone 20,22-monoacetonide, and *p*-hydroxyphenethyl *trans*-ferulate were isolated by vacuum column chromatography. In addition, *p*-hydroxyphenethyl *trans*-ferulate was the most active metabolite (IC_50_ = 19.24 ± 1.73 µM) [43].

The results obtained in this work suggested that, in the methanol-soluble fraction, there is a low presence of metabolites with inhibitory activity in αA and αG. In contrast, there is a major content of metabolites with inhibitory activity in DPP-IV. In addition, *Sida* spp. extracts were classified as non-toxic based on their LC_50_ values from a preliminary brine shrimp acute toxicity test (1.41 mg/mL and >2.00 mg/mL, for *S. rhombifolia* and *S. glabra*, respectively), which also were notably higher than the control colchicine (0.55 mg/mL). *A. salina* has been widely used as a model organism to test natural products’ toxicities due to its simplicity, cheapness, and correlation with the results obtained using acute oral toxicity assays in mice [31]. On the other hand, both crude extracts showed IC_50_ values with a relationship that supported the selective DPP-IV inhibition of *S. rhombifolia*. However, regarding effectiveness, higher IC_50_ values for DPP-IV were obtained (2.67 and 2.26 mg/mL for *S. rhombifolia* and *S. glabra*, respectively) compared with the reported IC_50_ of sitagliptin (7.33 × 10^−6^ mg/mL = 18 nM) [32,33]. This pattern indicated that chemical analysis must be carried out to identify potential bioactive compounds with safer and more effective activity profiles concerning the crude extracts.

Due to the biological activities exhibited by *Sida* species, several efforts have been made to characterize their chemical composition [77,78]. In this work, 22 specialized over-accumulated metabolites were tentatively identified in *S. rhombifolia*. In chemical–structural terms, 59% of the annotations corresponded to phenolic compounds (flavonoids, coumarins, phenylpropanoids, lignans, and stilbenoids), 27% for triterpenoids and steroids, and 15% for apocarotenoids, alkaloids, and indole derivatives. The tentative presence of phenolic compounds was particularly interesting since, according to the literature, these metabolites have been positioned as naturally occurring DPP-IV inhibitors [79]. For example, among more than 27 metabolites evaluated in vitro, resveratrol (IC_50_ = 0.6 ± 0.4 nM), luteolin (IC_50_ = 0.12 ± 0.01 µM), apigenin (IC_50_ = 0.14 ± 0.02 µM), and flavone (IC_50_ = 0.17 ± 0.01 µM) presented higher inhibitory activity than diprotin A (IC_50_ = 4.21 ± 2.01 µM), a known DPP-IV inhibitor with an Ile-Pro-Ile sequence. Additionally, through a molecular docking study, it was observed that resveratrol and flavone bound well into the S1, S2, and S3 DPP-IV subsites, whereas luteolin and apigenin could only bind into S2 and S3 subsites. For all compounds, hydrogen bonding was the main binding interaction with the active site of the enzyme [80]. Moreover, ecdysteroids have been tentatively identified in hydroalcoholic extracts of *S. rhombifolia* [81]. From those, 20-hydroxyecdysone and polypodine B stand out since they were identified in this work as some of the metabolites over-accumulated in *S. rhombifolia* compared with *S. glabra*. Ecdysteroid hormones, whose presence in plants protects against phytophagous insects, are a group of metabolites consistently identified in the *Sida* genus [82]. In previous studies, oral administration of 20-hydroxyecdysone decreased body fat and insulin resistance in C57BL/6J mice fed with a high-fat diet but did not show inhibitory activity in αG in vitro [43,83]. To our knowledge, except for ferulic acid (IC_50_ = 34.05 ± 0.29 mM) [84], the accumulated metabolites tentatively identified in *S. rhombifolia* have not been evaluated in vitro to determine their inhibitory activity on DPP-IV. On the other hand, corchoionoside C was inactive in αG (IC_50_ > 300 µM) [85], and kaempferitrin exhibited activity in αG (IC_50_ > 86.51 µM) but was inactive in αA (<20% inhibition at 500 µg/mL) [86]. *trans*-Tiliroside (IC_50_ = 2128 ± 63 µM) showed higher inhibition than acarbose (IC_50_ = 6561 ± 207 µM) in αG and has been inactive in αA [87]. Finally, ferulic acid has shown inhibitory activity in both enzymes (IC_50_ = 622 and 866 µg/mL for αA and αG, respectively) [88].

Based on the previous reports of phenolic compounds as novel DPP-IV inhibitors, a dereplication study was carried out for the first time for *S. glabra* in this work, reporting the presence of 29 phenolic compounds. Among those, rutin was the major metabolite (267.5 ± 3.7 µg/g). Also, 12 compounds were quantified for the first time in *S. rhombifolia*: 4-hydroxybenzoic acid, *trans*-cinnamic acid, (+)-catechin, apigenin, kaempferide, kaempferol-3-*O*-glucoside, kuromanin, luteolin, naringin, quercetin-3,4′-di-*O*-glucoside, quercetin-3-D-galactoside, and secoisolariciresinol. In preceding phenolic dereplication studies, 31 compounds were quantified in *S. rhombifolia*, of which isoquercetin was the major one (8.6 ± 1.12 mg/g extract) [73]. In contrast, vanillin showed the highest concentration in this work (17.5 ± 0.47 μg/g dry plant material). These findings contribute to the phytochemical knowledge about the *Sida* genus and support these CF species as new sources of phenolic compounds.

The comparative analysis of *S. rhombifolia* with *S. glabra* allowed us to identify naringin, vanillin, vanillic acid, kuromanin, and *trans*-cinnamic acid as metabolites accumulated in *S. rhombifolia*. It is essential to highlight that kuromanin has shown higher inhibition of DPP-IV compared to αG [89], naringin has shown inhibition of DPP-IV comparable to sitagliptin [90], and *trans*-cinnamic acid showed low inhibition of DPP-IV (4.4 ± 1.6%) at a concentration of 500 µM [91]. Finally, for vanillic acid and vanillin, the inhibitory activity of αA and αG has been evaluated [92,93], but no studies have been conducted on its activity on DPP-IV.

Within computer-assisted drug development (CADD), virtual screening is the most widely used tool since it allows systematic searches for novel small molecules with biological activity. With this aim, molecular docking is used to speed up the discovery of active compounds when the 3D structure of a molecular target is available [94,95]. This work’s ensemble molecular docking study allowed the selection of 10 natural products over-accumulated in the *S. rhombifolia* methanolic extract with high affinity for the active sites of DPP-IV, αA, and αG. From this subgroup, in addition to presenting lower BFE values than the reference drug sitagliptin, acanthoside D and *cis*-tiliroside were positioned as potential selective inhibitors of DPP-IV. The in silico study allowed us to determine that both metabolites established intermolecular interactions with amino acids relevant to the activity of DPP-IV: His 708 (catalytic triad of DPP-IV), Tyr 515, and Trp 595 (S1 and S2 extended, respectively). The interaction of a ligand with the amino acids in S1 and S2 is indispensable for the inhibitory activity in DPP-IV. However, the additional interaction with the S2 extended subsite has increased the potency of the enzymatic inhibition [51]. The combination of experimental and computational approaches has recently allowed the identification of compounds with selective inhibitory activity on molecular targets used in treating DMII. For example, by chemical synthesis, a spiroimidazolidinedione derivative was obtained, which presented higher inhibitory activity on aldose reductase (ALR2, IC_50_ = 0.47 µM) compared to aldehyde reductase (ALR1, IC_50_ = 60.1 µM). These enzymes share 65% homology, but the difference in their active sites is critical for selectivity. Through molecular docking studies, it was determined that the spiroimidazolidinedione derivative presented more affinity for ALR2 (−7.2 kcal/mol) in contrast to ALR1 (−5.8 kcal/mol). Also, this molecule established intermolecular interactions with Trp 111, Thr 113, and Cys 298, which are present in the active site of ALR2 [96]. On the other hand, anthocyanins have been isolated from blueberry, among which cyanidin-3-arabinoside showed the best inhibitory activity on protein tyrosine phosphatase 1B (IC_50_ = 8.91 ± 0.63 µM). Additionally, cyanidin-3-arabinoside exhibited lower inhibition on the closest T-cell protein tyrosine phosphatase (74% homology in their catalytic domain). A molecular docking study indicated that the difference between the amino acids around the active pockets of both enzymes could produce differences in their catalytic activity [97]. In that sense, evidence in the literature suggests that molecular docking currently allows obtaining information to propose protein–inhibitor interaction modes, which can generate plausible explanations for the experimental profiles of biological activity.

Acanthoside D, also named (−)-syringaresinol-di-β-D-glucoside, is a furofuranoid-type lignan described in different plants, including *Daphne feddei* and *D. giraldii* (Thymelaeaceae), which are related to the order level with *S. rhombifolia* (Malvales) [44,98]. This metabolite has exhibited significant inhibition of pulmonary inflammation in acute lung injury treated with lipopolysaccharides in mice by oral administration, as well as alleviation of liver injury induced by ischemia–reperfusion by inhibiting inflammatory cell infiltration [99,100]. Its hypoglycemic activity has not been experimentally evaluated. However, the related metabolite acanthoside B, also named (−)-syringaresinol-β-D-glucoside, has been inactive in in vitro enzyme inhibition assays in αG [101,102]. On the other hand, *cis*-tiliroside (flavonoid) has been described in the Malvaceae family (*Lasiopetalum macrophyllum*) and has reported inhibitory activity in αG (IC_50_ = 44.8 µM) [35,103]. As far as we know, acanthoside D and *cis*-tiliroside were identified for the first time in *S. rhombifolia* in this work. The results obtained about their in silico DPP-IV affinity, as well as the lack of experimental information on their inhibitory activity, places both metabolites as the best candidates for performing in vitro and in vivo biological activity assays in the future to evaluate them as selective and naturally occurring DPP-IV inhibitors.

## 4. Materials and Methods

### 4.1. Chemicals

The phosphate buffer pH = 7.2 (17202), 2-chloro-4-nitrophenyl-α-D-maltotrioside (93834, ≥96.0% for HPLC), acarbose (A898, ≥95% for HPLC), 4-nitrophenyl α-D-glucopyranoside (N1377, ≥99%), αA from porcine pancreas (A3176, Type VI-B, ≥five units/mg solid), αG from *S. cerevisiae* (G0660, ≥100 units/mg protein), and DPP-IV inhibitor screening kit (MAK203) were purchased from Sigma-Aldrich (St. Louis, MO, USA). The formic acid (28905, >99% for LC-MS) was purchased from Thermo Fisher Scientific (Rockford, IL, USA). The acetonitrile with 0.1% formic acid (14272, for LC-MS) and the methanol (34860, ≥99.9% for HPLC) were obtained from Honeywell (Muskegon, MI, USA). Water (9831-03, for LC-MS) was purchased from J. T. Baker (Phillipsburg, NJ, USA).

### 4.2. Rational Selection of the Cloud Forest Botanical Families with Potential Hypoglycemic Activity

#### 4.2.1. Cloud Forest Reference Study Site

The “Santuario del Bosque de Niebla” (SBN) is an ecological reserve of 30 hectares of CF belonging to the Institute of Ecology, A. C. (INECOL) in Xalapa, Veracruz, Mexico. It comprises diverse plant communities in different conservation stages, including tree, shrub, herbaceous, liana, and vine species. It is estimated that, in its extension (0.24% of the territory of the municipality of Xalapa), the SBN contains around 50% of the seed plants described for the region. According to recent updates on its biodiversity, 450 native angiosperms species have been found, distributed in 322 genera and 102 families, of which there is a record in the XAL Herbarium of INECOL [104]. This list was used as a reference to search for libraries of metabolites described for species belonging to the botanical families present in this CF site.

#### 4.2.2. Ligand-Based Virtual Screening

A metabolites library was built with molecules reported for 43 plants belonging to 13 families in the CF [104], which is available on the PMN platform [105]. The structures were visualized with Maestro (release 2022-2, Schrödinger LLC, New York, NY, USA), and, subsequently, the LigPrep tool was used to correct valence errors and define the stereochemistry. For each molecule, pharmacokinetic and physicochemical (ADME) properties were predicted with the Qikprop tool V 13.2128. The processed library was reduced through four filtering steps. The first step consisted of removing compounds with a molecular mass lower than 150 Da and higher than 500 Da, in order to mainly eliminate monosaccharides and nitrogenous bases (primary metabolites that are out of the scope of this study). In the second step, metabolites with atypical physicochemical properties compared to commercial oral drugs were removed. First, the drug-likeness of the metabolites was evaluated using the #stars calculated by QikProp. This parameter corresponds to those descriptors that fall outside of 95% of the values for known drugs. Metabolites with more than five stars were eliminated since it is a cut-off that indicates the molecule’s properties are similar to those drugs mostly used in a clinical setting [106]. In addition, molecules with more than two violations of Lipinsky’s rule of five (molecular mass < 500, cLogP < 5, and hydrogen bond donors and acceptors less than 5 and 10, respectively) and Jorgensen’s rule of three (predicted log S > −6, Pcaco > 30 nm/s, maximum number of primary metabolites = 6) were removed due to the fact that they exhibited properties associated with poor oral drug absorption [107,108]. In the third step, we selected molecules with functional groups present in phenolic compounds, amine-containing compounds, and terpenoids in order to create a link with the characteristic structures of specialized metabolites. Finally, the Tanimoto coefficient was calculated using the Platform for Unified Molecular Analysis (PUMA) to select molecules with potential hypoglycemic activity [109]. In this way, a Tanimoto coefficient ≥0.85 between two chemical structures is used as a cut-off for biological activity prediction [110]. A value of 100% was assigned to the total metabolites present in each group. Subsequently, matches were searched between the potentially active molecule groups and the original chemical compounds library. The families with the highest percentage of representativeness of the molecules of interest in each case were selected as the best candidates for their experimental study.

### 4.3. Plant Material Collection and Extract Preparation

The aerial parts of 12 species belonging to the potential hypoglycemic botanical families were collected in July 2022 from two CF reserves in Veracruz, Mexico: the SBN (Xalapa municipality, 19°30′50.845″ N, 96°56′31.99″ W) of INECOL and “La Martinica” (Banderilla municipality, 19°35′4.52″ N, 96°57′1.046″ W) (Table 6). All species were taxonomically identified by personnel from Herbarium XAL (INECOL). The plant material was cleaned and stored at −20 °C. The samples were freeze-dried (FreeZone 2.5, Labconco, Kansas City, MO, USA) and milled with a food processor (NBR-0801, Nutribullet, Los Angeles, CA, USA). The extraction was performed with methanol using an accelerated solvent extraction system (Dionex ASE 350, Thermo Fischer Scientific, Sunnyvale, CA, USA) [111]. The extraction temperature was 60 °C, and two static cycles of five minutes were used for each sample, with a rinse volume of 30% of the cell volume and nitrogen as carrier gas. The solvent was removed by rotatory evaporation (R-100, Büchi, Flawil, Switzerland) at 40 °C. The extraction yield was calculated on the dry weight of the crude drug (Table 6). The crude extracts were stored at −20 °C until use.

### 4.4. Enzymatic Inhibition Activity

#### 4.4.1. αA and αG Inhibition Assays

Inhibition assays for αA and αG were performed as previously described by our research group [112]. Briefly, for αA, all the reagents and extracts were prepared in a phosphate buffer solution (40 mM, pH = 6.8). In 96-well plates, 60 µL of the extract solution (1 mg/mL in the reaction mix) was mixed with 120 µL of 2-chloro-4-nitrophenyl-α-D-maltotrioside (0.35 mM) and incubated for five minutes at 35 °C. Then, 30 µL of αA (4 U/mL) was added, and the mix was incubated for ten minutes at 35 °C. Absorbance was measured at 405 nm at the beginning and end of the reaction (Multiskan FC, Thermo Fischer Scientific, Waltham, MA, USA). Acarbose (3 mM) was used as a positive control. On the other hand, for αG, the reagents and extracts were prepared in a phosphate buffer solution (100 mM, pH = 7.2). In 96-well plates, 20 µL of the extract solution (1 mg/mL in the reaction mix) was mixed with 100 µL of 4-nitrophenyl α-D-glucopyranoside (1 mM) and incubated for five minutes at 30 °C. Then, 100 µL de αG (0.005 mg/mL) was added, and the mix was incubated for 30 min at 30 °C. Absorbance was measured at 405 nm at the beginning and end of the reaction (Multiskan FC, Thermo Fischer Scientific, Waltham, MA, USA). Acarbose (30 mM) was used as a positive control. For both αA and αG, a reaction blank was included for each sample, and all assays were performed in triplicate. IC_50_ for selected extracts and acarbose was obtained from a concentration–response curve. The inhibition percentage (IP) was calculated with Equation (1), where A_N_ corresponds to the negative control absorbance (non-inhibited enzyme) and A_I_ to the absorbance in the presence of the inhibitor.
(1)$$IP=\left(\frac{A_{N}-A_{I}}{A_{N}}\right)\times 100$$

#### 4.4.2. DPP-IV Inhibition Assay

The in vitro inhibition of DPP-IV was evaluated with a commercial kit (see Section 4.1). All the reagents were prepared according to the manufacturer’s instructions, and the assays were carried out with some modifications. The extracts were prepared in the buffer solution (1 mg/mL in the reaction mix). Then, 12.5 µL of the extract solution was mixed with 25 µL of DPP-IV enzyme solution and incubated for 10 min at 37 °C. After, 12.5 µL of Gly-Pro-7-amido-4-methylcoumarin hydrobromide solution was added, and the fluorescence (λ_ex_ = 360/λ_em_ = 460 nm) was measured at 37 °C in kinetic mode for 30 min (Varioskan LUX, Thermo Fischer Scientific, Waltham, MA, USA). Sitagliptin was used as a positive control, and a reaction blank was included for each case. Assays were performed in triplicate. Fluorescence was plotted as a function of time, and its linear regression equation was obtained. IC_50_ for selected extracts was obtained from a concentration–response curve. The IP was calculated with Equation (2), where M_N_ corresponds to the slope of the linear regression equation of the negative control (non-inhibited enzyme) and M_I_ to the slope of the linear regression equation of the reaction in the presence of the inhibitor.
(2)$$IP=\left(\frac{M_{N}-M_{I}}{M_{N}}\right)\times 100$$

### 4.5. Acute Toxicity Assay with Artemia Salina (Brine Shrimp)

The culture conditions and acute toxicity test of selected plant extracts were carried out using a previously reported protocol [113]. One g/L of brine shrimp cysts (Ocean Star International, Snowville, UT, USA) were incubated for 48 h in artificial seawater (40 g sea salt/L, pH = 9.0) at 28 °C and a constant air supply. In 96-microwell plates, 100 µL of the extract solution prepared in artificial seawater with dimethyl sulfoxide (2% *v*/*v* initial) was mixed with 100 µL of the nauplius suspension (containing 13 ± 2 organisms). Each concentration was evaluated in triplicate, and a vehicle control was included. The plate was incubated at 28 °C for 24 h. After this, the dead nauplius (non-motile for 10 s) were counted in each well. Finally, 100 µL of methanol was added, and, 30 min later, the total number of nauplius was counted. LC_50_ was obtained from a concentration–response curve.

### 4.6. Untargeted Metabolomic Analysis

Crude extracts of selected plant species were dissolved in methanol (MS grade) at 50 mg/mL and filtered with a 0.2 µm PTFE membrane. Afterward, the analysis was performed on an ultra-high-performance liquid chromatography system (Acquity UPLC I Class, Waters, Milford, MA, USA) coupled with a high-resolution mass spectrometer (Q-TOF; Synapt G2-Si, Waters, Milford, MA, USA). The samples were separated on an Acquity BEH column (2.1 mm × 50 mm, 1.7 µm, C18, Waters, Milford, MA, USA) at 40 °C for the column oven and 15 °C for the autosampler. The chromatographic method consisted of an elution gradient with water (A) and acetonitrile (B), both acidified with formic acid (0.1% *v*/*v*), with an injection volume of 5 µL and a flow rate constant of 0.3 mL/min (Appendix A). Data management was performed with MassLynx (V. 4.1, Waters, Milford, MA, USA) and MarkerLynx (V 4.1, Waters, Milford, MA, USA). Exact mass values, retention times, and signal intensities (counts) were analyzed in the MetaboAnalyst 6.0 bioinformatic platform [114]. The tentative annotation of the over-accumulated selected metabolites was carried out by comparing the exact mass values obtained with the spectrometric data available in public spectral databases, such as FooDB V 1.0 [115], Lotus [116], PubChem [117] and MassBank [118].

### 4.7. Phenolics-Targeted Metabolomic Analysis

Thirty phenolic compounds were identified and quantified as previously described by our research group [119]. This targeted analysis was carried out in an ultra-high-performance liquid chromatography system (1290 Infinity series, Agilent Technologies, Santa Clara, CA, USA) coupled to a triple-quadrupole mass spectrometer (QqQ; 6460, Agilent Technologies, Santa Clara, CA, USA). The separation was performed on an Acquity BEH column (2.1 mm × 50 mm, 1.7 µm, C18, Waters, Milford, MA, USA) at 40 °C. The chromatographic method consisted of an elution gradient with water (A) and acetonitrile (B), both acidified (0.1% *v*/*v*) with formic acid, with an injection volume of 2 µL and a flow rate constant of 0.3 mL/min. The acquisition method consisted of Dynamic Multiple Reaction Monitoring (dMRM) (Appendix A). The identity of all analytes was confirmed by co-elution with authentic standards under the same analytical conditions. For the quantification, calibration curves in a concentration range of 0.5–19 µM were used (r^2^ ≥ 0.97 for each case). Data management was performed with MassHunter Workstation V B.06.00 (Agilent Technologies, Santa Clara, CA, USA) and MetaboAnalyst 6.0. The results for the quantification of each analyte were expressed as mean (µg/g dried material) ± standard deviation.

### 4.8. Ensemble Molecular Docking

#### 4.8.1. Metabolites Library Preparation

The tentatively annotated metabolites and the phenolics dereplicated in the active extracts were selected as candidates for potential selective inhibitory activity. The 3D structures in sdf format for all the interest metabolites were retrieved from PubChem and prepared with the molecular visualization software Maestro (release 2022-2, Schrödinger LLC, New York, NY, USA). Additionally, the 3D structures of the drugs acarbose, sitagliptin, and metformin were included to act as controls in the in silico study. All the structures were processed with the protein preparation wizard tool (OPLS-AA forcefield), minimized with the 3D builder tool, and exported in mol2 format.

#### 4.8.2. Molecular Targets Preparation

The structures of the co-crystallized enzyme–inhibitor complexes of human pancreas αA (4W93), *S. cerevisiae* αG (3A47), and human DPP-IV (3W2T) were retrieved from the RCSB Protein Data Bank database. The structures were visualized in Maestro Suite V 12.4.072 (OPLS-AA forcefield) and processed with the protein preparation wizard tool for the identification of the atom types, bond distances, addition of hydrogens, creation of disulfide bonds, and generation of ionization states corresponding to pH = 7 using the Propka method. Molecular dynamics simulations (MDs) were performed using pmemd.cuda from the Amber20 package to obtain a description of the flexibility of each molecular target [120]. The parameters and topologies describing the enzyme and the inhibitors come from the ff14SB and GAFF force field (for proteins and organic molecules) within the Amber20 simulation package. The enzyme–inhibitor complexes were solvated in an orthorhombic chemical space of 15 Å per side using the TIP3P water model (optimized for solvated proteins) and 150 mM NaCl. The number of particles in each system was as follows: DPP-IV = 345,125 atoms (1491 amino acids, 106,716 explicit water molecules), αA = 325,392 atoms (495 amino acids, 39,120 explicit water molecules), and αG = 132,773 atoms (586 amino acids, 41,039 water). The initial minimization (combined steepest descendant and conjugated gradient scheme) and simulated annealing were performed with an NvT ensemble. The annealing consisted of six steps: 300 ps at 10 K, 1000 ps at 100 K, 1000 ps at 300 K, 300 ps at 400 K, 500 ps at 400 K, and, finally, 2000 ps at 300 K. The system was then equilibrated for 7500 ps and conventional MDs was performed for 250 ns, both with an NpT ensemble (p = 1 atm, Monte Carlo Barostat = 2). Long-range electrostatic interactions were calculated with the Particle Mesh Ewald method, and an integration time of 2 fs was used. Production calculations were performed for 250 ns of Gaussian Accelerated MDs (sigma0 = 6.0 and sigma0D = 6.0).

#### 4.8.3. Ensemble Molecular Docking Calculations

From the trajectories of MDs, six representative conformers of each target were extracted and docked with the molecules in the metabolite library using the GPU version of AutoDock Vina (GNINA V 1.1) [121] and the Vinardo scoring function (Monte Carlo-Metropolis exhaustive search method of 96) [122]. The chemical search space consisted of an orthorhombic system of 30 Å per side and was situated in the Cartesian coordinates: DPP-IV (83, 78, 82), αA (67, 42, 48), and αG (68, 52, 55). All BFE values in kcal/mol were reclassified using a neural network method, which is included in the redock_default2018 set in the GNINA program. The results were expressed as the mean of BFE (kcal/mol) ± standard deviation of the values calculated between each conformer (*n* = 6) and the binding modes of the ligands (*n* = 5) in the database (GraphPad Prism V 10.2.3, GraphPad Software, San Diego, CA, USA).

### 4.9. Statistical Analysis

#### 4.9.1. Enzyme Inhibitory Activity and Acute Toxicity Evaluation

The results of the inhibitory enzymatic activity were expressed as a mean with a standard deviation. Data normality was evaluated through the Shapiro–Wilk test. After this, a one-way analysis of variance and a post hoc Dunnett test was performed to establish a statistically significant difference compared to the positive controls (*p ≤* 0.05). IC_50_ values were determined by non-linear regression analysis of the log_10_[concentration]–response curves. LC_50_ values in the brine shrimp toxicity test were calculated using a logistic regression analysis. All the analyses mentioned in this section used GraphPad Prism V 10.2.3 for Windows (GraphPad Software, San Diego, CA, USA).

#### 4.9.2. Metabolomic Informatic Analysis

Data analysis for chemical profiling of the selected species was performed in the Metaboanalyst 6.0 platform [114]. For the untargeted analysis, analytical intensity values of the spectrometric features (rt-*m*/*z*) were used, and the data were square root transformed and ranged scaled. The hierarchical clustering heatmap was performed using the Euclidean distance and Ward clustering methods. The volcano plot was made with a fold change threshold ≥ 2 (*S. rhombifolia*/*S. glabra*) and an FDR value of 0.05. Finally, an enrichment analysis of metabolic pathways was performed using the Mummichog algorithm, the KEGG database, and the *A. thaliana* library (dicots). For the targeted analysis, the quantification values of the phenolic compounds (µg/g dried material) were used, and the data were Log_10_ transformed and mean centering scaled. The volcano plot was made with a fold change threshold ≥ 2 (*S. rhombifolia*/*S. glabra*) and an FDR value of 0.05.

## 5. Conclusions

Bioactive plants from CF Veracruz were identified by applying the chemotaxonomic computer-assisted selection strategy designed and validated in this work. The in silico prediction of the hypoglycemic activity at the botanical family level allowed the inclusion of species not used in traditional Mexican medicine to treat DM. The results obtained in evaluating the hypoglycemic activity contribute to the knowledge of the therapeutic properties of CF species in Mexico and support this ecosystem as a reservoir of plants with pharmacological potential. The evaluated species are promising sources of specialized metabolites with inhibitory activity on the enzymes αA, αG, and DPP-IV, among which *S. rhombifolia* stands out for its selective DPP-IV inhibition profile. The experimental strategy allowed us to identify 27 natural products to which the biological activity of *S. rhombifolia* could be attributed. Through the ensemble molecular docking study, an additional filtering process was incorporated, and acanthoside D and *cis*-tiliroside were selected as the best candidates to continue with the discovery of potential natural hypoglycemic agents with selective DPP-IV inhibition. Noteworthily, the overall workflow used in this work could be applied to identify plants with different biological activities by including other reference drugs in the LBVS and relevant biological assays in order to confirm the in silico predictions. Finally, this approach is proposed as a novel tool for pharmacological bioprospecting that could accelerate the discovery of bioactive compounds in biodiversity hotspots in which the lack of ethnopharmacological information limits the use of conventional criteria.

## Figures and Tables

**Figure 1 ijms-25-10881-f001:**
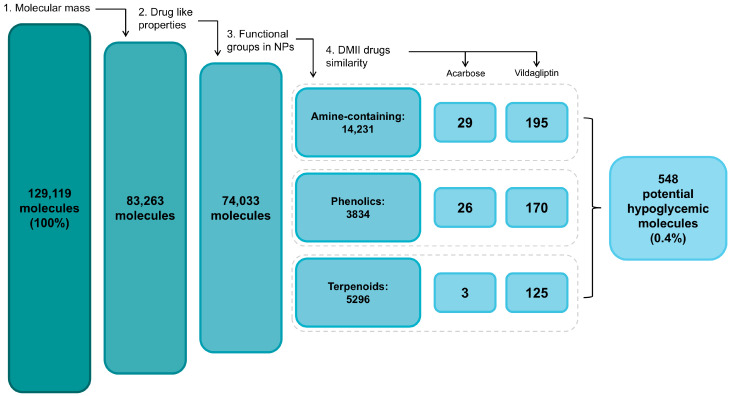
Filtering steps in LBVS to select molecules with potential hypoglycemic activity. NPs = natural products.

**Figure 2 ijms-25-10881-f002:**
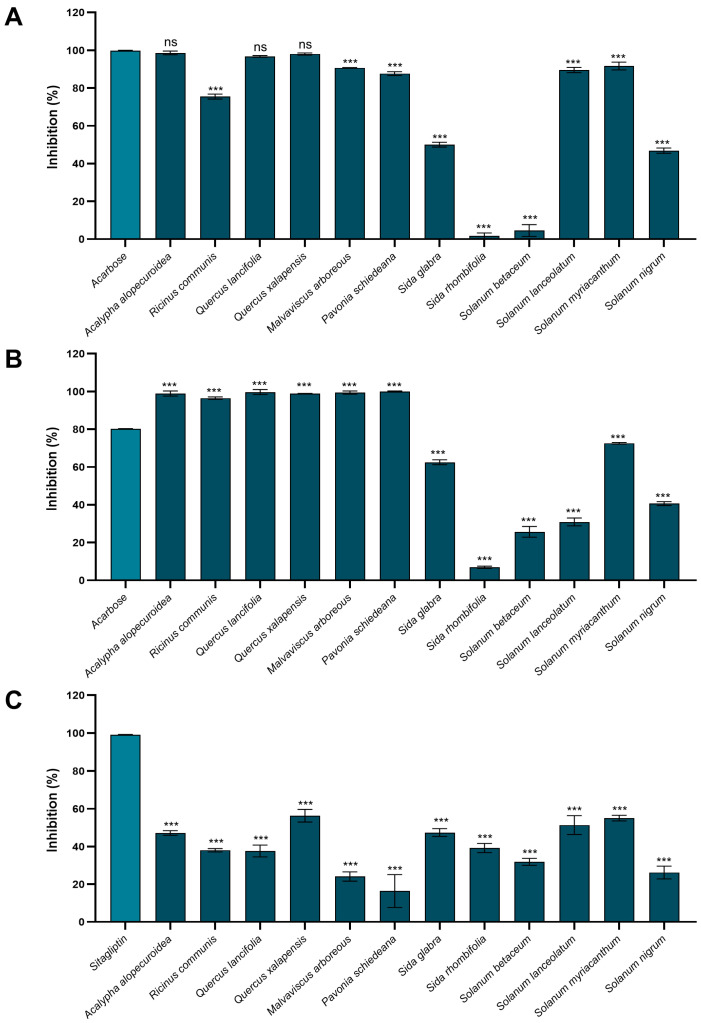
Percentages of inhibition of (**A**) αA, (**B**) αG, and (**C**) DPP-IV of the methanolic crude extracts of the CF plant species (1 mg/mL). *** Statistically significant difference (*p* ≤ 0.05) and ns = no statistically significant difference (*p* > 0.05) compared to positive controls: acarbose (αA = 3 mM, αG = 30 mM), and sitagliptin (DPP-IV), based on one-way analysis of variance with a post hoc Dunnett test. Results are expressed as mean (*n* = 3) ± standard deviation.

**Figure 3 ijms-25-10881-f003:**
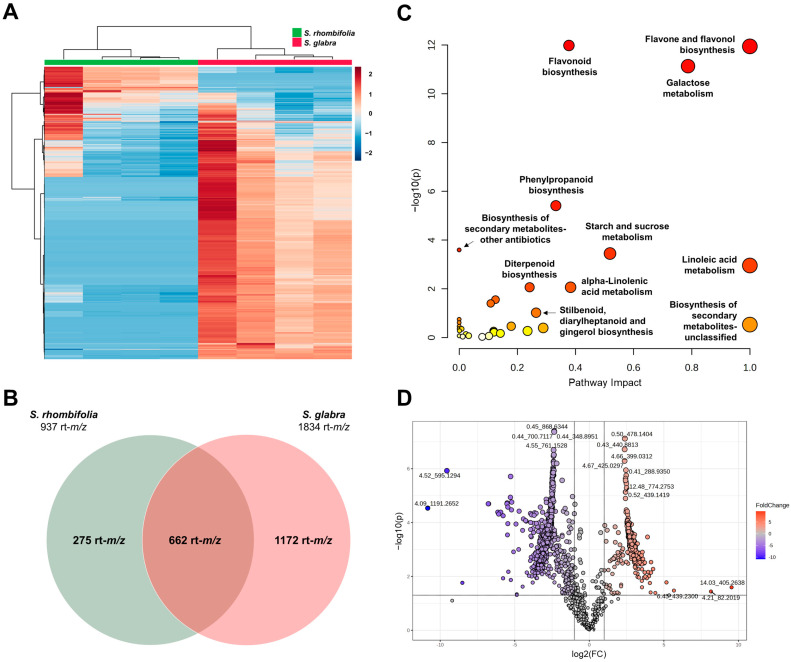
Untargeted metabolomic analysis of *S. rhombifolia* and *S. glabra*. (**A**) Hierarchical clustering heatmap showing the comparison of chemical profiles (generated using the Euclidean and Ward methods for distance estimation and as a clustering algorithm, respectively). Each row corresponds to a unique rt-*m*/*z* feature. The intensity comparison is shown by the *z*-score (red and blue colors mean higher and lower intensities, respectively, among samples). (**B**) Venn diagram with the distribution of rt-*m*/*z* features of both plant species. (**C**) Functional analysis showing the most impacted and significant metabolic pathways identified with the rt-*m*/*z* features detected in *S. rhombifolia* and *S. glabra* (generated using the Mummichog algorithm in MetaboAnalyst 6.0) with the *Arabidopsis thaliana* library deposited in KEGG database. Red to yellow color scale is related to the significant pathway impact [−log10(*p*)] value. (**D**) Volcano plot of the comparison of the global chemical profiles of *S. rhombifolia*/*S. glabra* (fold change ≥ 2 and *p* < 0.05). Each circle corresponds to a rt-*m*/*z* feature. Red and blue colors mean significant (*t*-test) positive and negative logarithmic fold change values, respectively.

**Figure 4 ijms-25-10881-f004:**
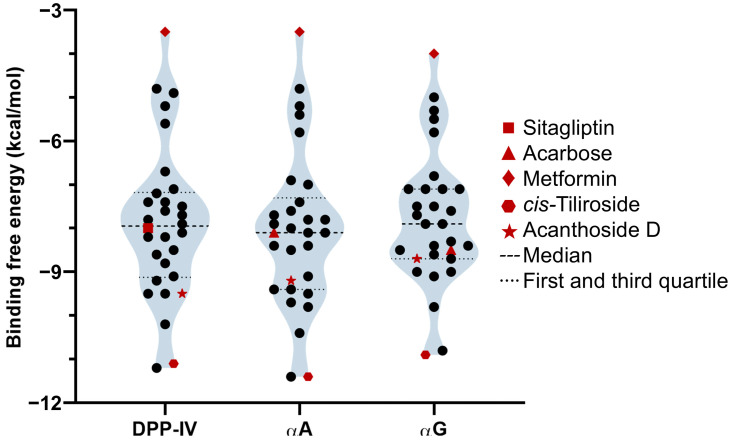
Distribution of ensemble molecular docking scores obtained for the over-accumulated metabolites of *S. rhombifolia* and the DMII molecular targets. Docking scores (BFE) were expressed as the mean of the replicates (binding modes) of each compound (*n* = 5) and the conformers of the enzyme (*n* = 6). Every symbol represents a different metabolite.

**Figure 5 ijms-25-10881-f005:**
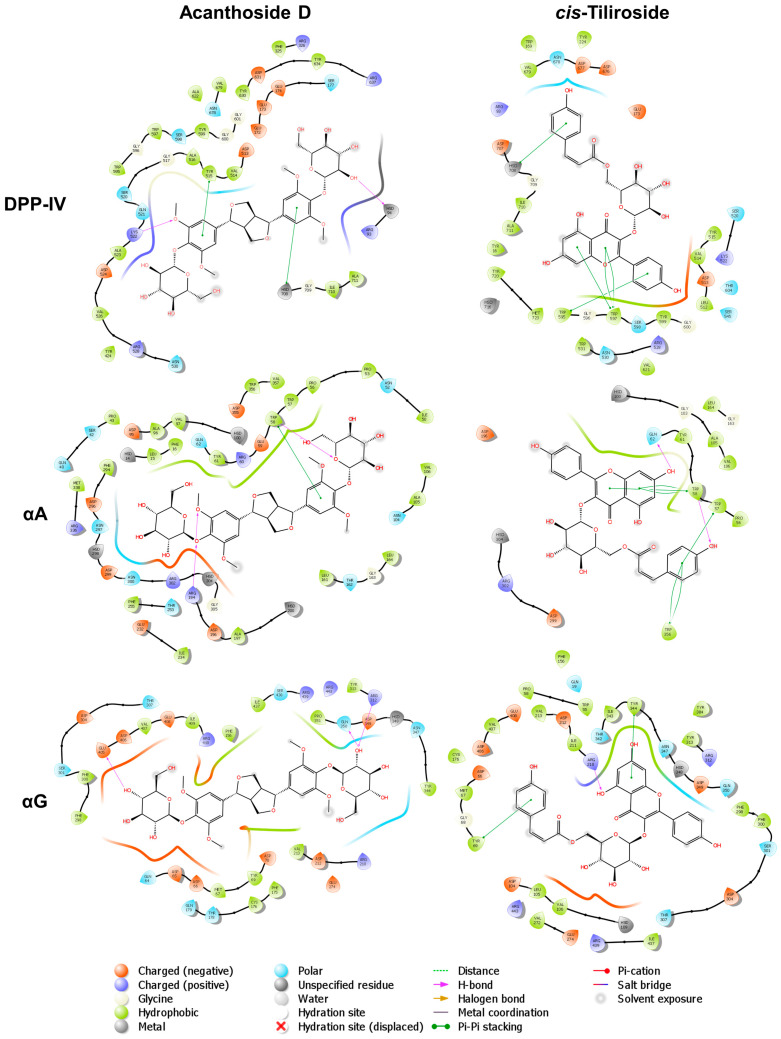
Ligand interaction diagrams of acanthoside D (**left**) and *cis*-tiliroside (**right**) vs. DMII molecular targets (DPP-IV, αA and αG). All interactions were calculated around a 6 Å distance cut-off.

**Table 1 ijms-25-10881-t001:** Percentage of the potential hypoglycemic metabolites identified in the selected CF botanical families using an LBVS.

	Amine-Containing TC ≥ 0.85 = 29 (A)	Amine-Containing TC ≥ 0.85 = 195 (V)	Phenolics TC ≥ 0.85 = 26 (A)	Phenolics TC ≥ 0.85 = 170 (V)	Terpenoids TC ≥ 0.85 = 3 (A)	Terpenoids TC ≥ 0.85 = 125 (V)
1	Brassicaceae (79%)	Brassicaceae (64%)	Malvaceae (69%)	Euphorbiaceae (63%)	Euphorbiaceae (100%)	Fagaceae (72%)
2	Fagaceae (38%)	Fagaceae (53%)	Solanaceae (65%)	Fagaceae (62%)	Fagaceae (100%)	Solanaceae (64%)
3	Euphorbiaceae (34%)	Euphorbiaceae (51%)	Anacardiaceae (62%)	Solanaceae (55%)	Brassicaceae (100%)	Brassicaceae (63%)
4	Solanaceae (28%)	Solanaceae (47%)	Amaranthaceae (58%)	Myrtaceae (44%)	Solanaceae (67%)	Euphorbiaceae (57%)
5	Asteraceae (21%)	Amaranthaceae (43%)	Fagaceae (54%)	Amaranthaceae (42%)	Amaranthaceae (67%)	Malvaceae (56%)
6	Malvaceae (21%)	Malvaceae (40%)	Euphorbiaceae (54%)	Malvaceae (42%)	Malvaceae (67%)	Rutaceae (54%)
7	Rutaceae (21%)	Rutaceae (39%)	Myrtaceae (54%)	Rutaceae (40%)	Rutaceae (67%)	Myrtaceae (49%)
8	Amaranthaceae (17%)	Asteraceae (38%)	Brassicaceae (50%)	Anacardiaceae (39%)	Anacardiaceae (67%)	Amaranthaceae (47%)
9	Anacardiaceae (17%)	Myrtaceae (36%)	Asteraceae (46%)	Brassicaceae (39%)	Asteraceae (67%)	Anacardiaceae (46%)
10	Cucurbitaceae (17%)	Cucurbitaceae (35%)	Rutaceae (38%)	Asteraceae (37%)	Cucurbitaceae (67%)	Cucurbitaceae (44%)
11	Dioscoreaceae (17%)	Anacardiaceae (34%)	Cucurbitaceae (38%)	Cucurbitaceae (35%)	Myrtaceae (0%)	Asteraceae (43%)
12	Myrtaceae (17%)	Dioscoreaceae (30%)	Dioscoreaceae (35%)	Dioscoreaceae (34%)	Dioscoreaceae (0%)	Dioscoreaceae (35%)
13	Araceae (14%)	Araceae (29%)	Araceae (31%)	Araceae (23%)	Araceae (0%)	Araceae (27%)

Molecules in each group with a Tanimoto coefficient (TC) ≥ 0.85 were calculated concerning acarbose (A) and vildagliptin (V).

**Table 2 ijms-25-10881-t002:** IC_50_ in enzymatic inhibition assays and LC_50_ in brine shrimp toxicity assay of *S. rhombifolia* and *S. glabra* extracts.

Sample	IC_50_ (mg/mL)			LC_50_ (mg/mL)
	αA	αG	DPP-IV	
*S. glabra*	1.27	0.33	2.26	>2.00
*S. rhombifolia*	>4.00	>4.00	2.67	1.41
Positive control	0.003 ^a^	2.88 ^a^	7.33 × 10^−6 b^ [32,33]	0.55 ^c^

^a^ Acarbose, ^b^ sitagliptin, ^c^ colchicine.

**Table 3 ijms-25-10881-t003:** Tentatively identified specialized metabolites over-accumulated in *S. rhombifolia* compared to *S. glabra*.

Fold Change	*p* Value	Retention Time (min)	Exact Mass	Name	Metabolite Type	Adduct	Mass Error (ppm)	Plant Species
11.99	1.1 × 10^−2^	4.16	409.1822	Corchoionoside C	Apocarotenoids	[M+Na]^+^	−4.0	*Corchorus olitorius* * [34]
11.31	1.1 × 10^−2^	6.79	595.1442	*cis*-Tiliroside	Flavonoids	[M+H]^+^	−1.6	*Lasiopetalum macrophyllum* * [35]
8.01	4.1 × 10^−4^	4.41	577.1547	Kaempferitrin	Flavonoids	[M−H]^−^	−1.8	*Tilia tomentosa* * [36]
7.40	8.7 × 10^−5^	7.82	385.0916	Cleomiscosin A	Coumarins	[M−H]^−^	−1.9	*Hibiscus syriacus* * [37]
7.25	4.4 × 10^−5^	4.44	333.1091	Cappariloside A	Small peptides	[M−H]^−^	1.3	*Capparis spinosa* [38]
7.18	8.8 × 10^−3^	6.55	617.1265	*trans*-Tiliroside	Flavonoids	[M+Na]^+^	−1.0	*Lasiopetalum macrophyllum* * [35]
7.01	9.0 × 10^−5^	6.66	385.0924	Cleomiscosin B	Coumarins	[M−H]^−^	0.1	*Hibiscus syriacus* * [37]
7.01	8.9 × 10^−5^	3.88	741.1872	Depressonol A	Flavonoids	[M−H]^−^	−0.8	*Corchorus depressus* * [39]
6.98	6.2 × 10^−5^	5.48	667.2601	Isolimonic acid glucoside **	Triterpenoids	[M−H]^−^	−0.1	*Citrus aurantium* [40]
6.98	6.2 × 10^−5^	5.48	667.2601	Ichangic acid 17-β-D-glucopyranoside **	Triterpenoids	[M−H]^−^	−0.1	*Citrus aurantium* [40]
6.89	3.1 × 10^−5^	5.85	371.1350	Citrusin E	Phenylpropanoids	[M−H]^−^	2.1	*Citrus limon* [41]
6.83	7.8 × 10^−5^	4.85	419.1338	Rhaponticin	Stilbenoids	[M−H]^−^	−1.0	*Eucalyptus rubida* [42]
6.81	7.0 × 10^−5^	5.80	515.2773	20-Hydroxyecdysone	Steroids	[M+Cl]^−^	−0.5	*Sida rhombifolia* * [43]
6.80	5.9 × 10^−5^	4.27	741.2592	Acanthoside D	Lignans	[M−H]^−^	−1.9	*Daphne feddei* [44]
6.67	7.1 × 10^−5^	7.63	415.1025	Cleomiscosin D	Coumarins	[M−H]^−^	−1.0	*Hibiscus syriacus* * [37]
6.62	6.6 × 10^−5^	4.12	461.1094	Thermopsoside	Flavonoids	[M−H]^−^	2.2	*Daphne feddei* [45]
6.54	3.3 × 10^−5^	3.39	193.0497	Ferulic acid	Phenylpropanoids	[M−H]^−^	−2.0	*Sida acuta* * [46]
6.51	8.1 × 10^−5^	4.26	385.1135	1-*O*-Sinapoyl-β-D-glucose	Phenylpropanoids	[M−H]^−^	0.1	*Arabidopsis thaliana* [47]
5.47	5.0 × 10^−7^	13.59	612.2245	Neoacrimarine E	Alkaloids	[M−H]^−^	1.9	*Citrus paradisi/C. tangerina* [48]
3.69	9.0 × 10^−4^	4.56	531.2717	Polypodine B	Steroids	[M+Cl]^−^	−1.5	*Sida szechuensis* * [49]
2.67	3.4 × 10^−2^	12.49	699.3564	Corchorusoside D **	Steroids	[M+H]^+^	−4.0	*Corchorus olitorius* * [50]
2.67	3.4 × 10^−2^	12.49	699.3564	Corchorusoside A **	Steroids	[M+H]^+^	−4.0	*Corchorus olitorius* * [50]

* Reported in species from the Malvaceae family, ** undistinguished isomers.

**Table 4 ijms-25-10881-t004:** Dereplicated phenolics by UPLC-MS/MS in the crude extracts of *Sida* species.

Metabolite Type	Name	SR *	SG *	SR/SG FC	SR/SG *p* Value
Phenolic acids	4-Hydroxybenzoic acid	4.35 ± 0.46	2.63 ± 0.05	---	---
	Ellagic acid	1.21 ± 0.08	5.48 ± 0.38	0.221	1.84 × 10^−7^
	Gallic acid	0.07 ± 0.02	0.25 ± 0.01	0.284	3.86 × 10^−5^
	Protocatechuic acid	0.49 ± 0.05	1.55 ± 0.04	0.317	5.66 × 10^−7^
	Salicylic acid	12.54 ± 0.54	20.83 ± 0.97	---	---
	Vanillic acid	11.06 ± 0.09	3.51 ± 0.09	3.153	3.00 × 10^−10^
	Vanillin	17.50 ± 0.47	6.94 ± 0.06	2.523	5.05 × 10^−9^
Phenylpropanoids	4-Coumaric acid	1.78 ± 0.14	1.47 ± 0.03	---	---
	Caffeic acid	0.46 ± 0.04	1.28 ± 0.01	0.360	1.81 × 10^−7^
	Chlorogenic acid	0.30 ± 0.03	107.72 ± 1.29	0.003	2.29 × 10^−12^
	Ferulic acid	1.75 ± 0.04	1.03 ± 0.01	---	---
	Sinapic acid	0.00	0.18 ± 0.02	0.176	2.68 × 10^−7^
	*trans*-Cinnamic acid	0.28 ± 0.01	0.13 ± 0.00	2.091	3.41 × 10^−8^
Flavonoids	(−)-Epicatechin	0.00	2.60 ± 0.04	0.196	3.71 × 10^−11^
	(+)-Catechin	0.02 ± 0.01 **	4.49 ± 0.05	0.005	7.82 × 10^−10^
	Apigenin	0.02 ± 0.00 **	0.16 ± 0.00	0.153	5.05 × 10^−9^
	Kaempferide	0.00	0.84 ± 0.20	0.143	3.21 × 10^−5^
Flavonoids	Kaempferol	0.00	1.83 ± 0.07	0.190	2.85 × 10^−9^
	Kaempferol-3-*O*-glucoside	13.50 ± 0.57	48.34 ± 0.29	0.279	1.40 × 10^−9^
	Kuromanin	0.10 ± 0.01 **	0.01 ± 0.00 **	10.732	1.29 × 10^−6^
	Luteolin	0.05 ± 0.01 **	0.07 ± 0.00	---	---
	Naringin	0.53 ± 0.03	0.00	5.338	1.68 × 10^−8^
	Quercetin	0.00	0.50 ± 0.06	0.171	6.42 × 10^−7^
	Quercetin-3,4′-di-*O*-glucoside	0.07 ± 0.02 **	3.57 ± 0.08	0.021	1.41 × 10^−8^
	Quercetin-3-D-galactoside	0.84 ± 0.02	8.38 ± 0.15	0.100	3.11 × 10^−11^
	Quercetin-3-glucoside	1.56 ± 0.03	90.32 ± 1.37	0.017	2.29 × 10^−12^
	Rutin	3.36 ± 0.18	267.47 ± 3.72	0.013	2.71 × 10^−12^
Coumarins	Umbelliferone	0.04 ± 0.00	0.05 ± 0.00	---	---
Lignans	Secoisolariciresinol	0.33 ± 0.05	0.79 ± 0.09	0.41	1.06 × 10^−4^
Amino acids	Phenylalanine	14.77 ± 0.38	19.95 ± 0.37	---	---

SR = *S. rhombifolia*, SG = *S. glabra*, FC = fold change, and --- = no significant differences (*p* > 0.05). * Data are expressed as μg of compound/g dry material of the mean (*n* = 4) ± standard deviation. ** Calculated below the limit of quantification.

**Table 5 ijms-25-10881-t005:** Intermolecular interactions for the ten metabolites from *S. rhombifolia* with the lowest FBE values vs. the DMII molecular targets.

DPP-IV (PDB ID 3W2T)			
Metabolite	BFE ** (kcal/mol)	Hydrogen Bond	Pi–Pi Stacking
Acanthoside D *	−9.5 ± 0.37	His 94, **Lys 522**	**Tyr 515**, **His 708**
*cis*-Tiliroside *	−11.1 ± 0.59	-	**His 708, Trp 597, Trp 595**
Corchorusoside D *	−8.8 ± 0.33	**Tyr 515**, **Arg 326**, His 94	-
Depressonol A *	−10.2 ± 0.47	Glu 174, Gln 521, Tyr 424, Asp 524, Tyr 638	**Phe 325**
Kaempferitrin *	−9.5 ± 0.58	-	**Trp 595**, **Trp 597**
Kuromanin	−9.1 ± 0.54	-	Trp 595, Trp 597
Naringin *	−9.5 ± 0.37	Asp 524, **Arg 93**, Asp 513, Val 514	**Trp 597**
Rhaponticin *	−8.6 ± 0.51	Asp 513	**Trp 597**
Thermopsoside *	−9.2 ± 0.43	Gly 709	**Trp 595**, **Trp 597**
*trans*-Tiliroside *	−11.2 ± 0.57	Gln 521	**Phe 325**, **Trp 595**
Sitagliptin	−8.0 ± 0.25	**Asn 678**, **Tyr 630**	**Tyr 634**
Metformin	−3.5 ± 0.17	-	-
**αA (PDB ID 4W93)**			
**Metabolite**	**BFE ** (kcal/mol)**	**Hydrogen bond**	**Pi–Pi stacking**
Acanthoside D	−9.2 ± 0.33	Arg 194, Trp 58	Trp 58
*cis*-Tiliroside	−11.4 ± 0.45	Gln 62, Trp 58	Trp 58, Trp 57, Trp 356
Corchorusoside D	−9.1 ± 0.56	Thr 162, Asp 146	-
Depressonol A *	−10.4 ± 0.21	Gln 62, **Asp 196**, Ala 105, Thr 162	Trp 58
Kaempferitrin	−9.8 ± 0.68	Arg 194, Asp 355	Trp 57, Trp 58
Kuromanin	−9.4 ± 0.21	-	Trp 58
Naringin *	−9.5 ± 0.34	Arg 194, **Asp 196**, **Glu 232**	Trp 58
Rhaponticin	−9.4 ± 0.60	Gln 62, Arg 194	Trp 58
Thermopsoside *	−9.7 ± 0.78	**Glu 232**, **Asp 299**	Trp 58
*trans*-Tiliroside *	−11.4 ± 0.30	**Glu 232**, Ile 234, Tyr 61	His 200, Trp 57, Tyr 61
Acarbose	−8.1 ± 0.18	**Asp 299**, His 14, Gln 40, Gln 62, **Asp 196**	-
Metformin	−3.5 ± 0.30	Arg 194	-
**αG (3A47)**			
**Metabolite**	**BFE ** (kcal/mol)**	**Hydrogen bond**	**Pi–Pi stacking**
Acanthoside D	−8.7 ± 0.81	Glu 405, Gln 350, Arg 312	-
*cis*-Tiliroside	−10.9 ± 1.54	Arg 210	Tyr 69, Tyr 344
Corchorusoside D *	−8.4 ± 0.90	Arg 210, **Asp 212**	-
Depressonol A	−9.8 ± 1.53	Asn 347	Tyr 69
Kaempferitrin	−8.6 ± 1.29	Asn 412, Asn 411, Phe 156, Arg 312	Phe 300
Kuromanin	−8.7 ± 0.90	Gln 276	-
Naringin *	−9.1 ± 1.12	**Glu 274**, Arg 443	Tyr 69
Rhaponticin *	−9.0 ± 0.81	Arg 210, **Glu 274**, Tyr 155, Asp 304, Thr 307	Phe 175, Tyr 155
Thermopsoside *	−9.0 ± 0.71	**Glu 274**, **Asp 212**, Asn 347, Tyr 344	Phe 300
*trans*-Tiliroside	−10.8 ± 1.60	Arg 443	Tyr 69
Acarbose	−8.5 ± 0.41	Gln 276, **Glu 274**, Asp 66, **Asp 349**	-
Metformin	−4.0 ± 0.27	Gly 68, **Asp 212**	-

* Metabolites that interacted with relevant amino acids for the inhibitory activity. Key amino acids are shown in bold. ** BFE was expressed as the mean of the binding modes of each compound (*n* = 5) and the conformers of each enzyme (*n* = 6) ± standard deviation.

**Table 6 ijms-25-10881-t006:** Cloud forest plants targeted for hypoglycemic activity evaluation.

Botanical Family	Specie	Study Site	Voucher	Extraction Yield *
Euphorbiaceae	*Acalypha alopecuroidea* Jacq.	SBN	XAL0151034	23.8%
Euphorbiaceae	*Ricinus communis* L.	SBN	XAL0151031	29.5%
Fagaceae	*Quercus lancifolia* Schltdl. and Cham.	SBN	XAL0151032	19.9%
Fagaceae	*Quercus xalapensis* Bonpl.	SBN	XAL0151041	24.1%
Malvaceae	*Malvaviscus arboreous* Cav.	SBN	XAL0151030	14.0%
Malvaceae	*Pavonia schiedeana* Steud.	SBN	XAL0151035	16.1%
Malvaceae	*Sida glabra* Mill.	LM	XAL0151040	7.3%
Malvaceae	*Sida rhombifolia* L.	SBN	XAL0151025	9.9%
Solanaceae	*Solanum* betaceum Cav.	SBN	XAL0151027	19.8%
Solanaceae	*Solanum lanceolatum* Cav.	LM	XAL0151039	24.2%
Solanaceae	*Solanum myriacanthum* Dunal	SBN	XAL0151042	27.6%
Solanaceae	*Solanum nigrum* L.	LM	XAL0151037	12.3%

SBN = Santuario del Bosque de Niebla, LM = La Martinica. * The extraction yield was calculated on the dry weight of the crude drug (dried plant material).

## Data Availability

The original contributions presented in this study are included in the article/Appendix A. Further inquiries can be directed to the corresponding authors.

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
