# Peer review of "Targeting Hypoglycemic Natural Products from the Cloud Forest Plants Using Chemotaxonomic Computer-Assisted Selection"

_ijms, 2024, doi:10.3390/ijms252010881_

Round 1

Reviewer 1 Report

Comments and Suggestions for Authors

The manuscript titled Targeting hypoglycemic natural products from cloud forest plants using chemotaxonomic computer-assisted selection” highlights an innovative approach by using chemotaxonomic and computational methods to target plant species from the cloud forest with potential hypoglycemic activity. The work significantly contributes to the discovery of natural products that can inhibit enzymes related to diabetes management, specifically dipeptidyl peptidase IV (DPP-IV), α-amylase, and α-glucosidase. The identification of Sida rhombifolia as a promising source of DPP-IV inhibitors, along with a detailed analysis of its metabolomic profile, presents new avenues for the development of natural antidiabetic treatments. I believe that with minor revisions, this manuscript can make a valuable addition to the literature, and I recommend it for publication.

Minor Comments for Authors:

  1. Abstract and Introduction:
    • Consider revising the first sentence of the abstract to emphasize the pharmacological importance of cloud forest plants. For example, replace "The cloud forest (CF) is a highly biodiverse ecosystem scarcely studied in the pharmacological context" with something like "The cloud forest (CF), a biodiversity hotspot, holds untapped potential for discovering pharmacologically active compounds."
  2. Materials and Methods:
    • The description of the ligand-based virtual screening (LBVS) process could benefit from more detail on how the final selection criteria were determined. Providing clear reasoning for the thresholds used in each filtering step would enhance reproducibility.
  3. Results – Metabolomic Analysis:
    • The metabolomic analysis presented in Figure 3 is very informative. However, for greater clarity, you could include brief explanations within the figure legend to guide the reader through the key findings highlighted by the heatmap, Venn diagram, and volcano plot.
    • The untargeted metabolomic analysis, particularly the identification of metabolites in Sida rhombifolia and S. glabra, is a significant result. However, further validation of the biological activity of key identified compounds, such as acanthoside D and cis-tiliroside, would strengthen the conclusions drawn from the study. While the docking results are promising, experimental validation (e.g., in vitro or in vivo enzyme inhibition assays) would provide more direct evidence of their efficacy as selective DPP-IV inhibitors. Incorporating such data would give a more robust basis for the claim that these metabolites could serve as potential therapeutic agents for diabetes treatment.
    • The enzymatic inhibition profiles of the twelve species collected from the cloud forest, particularly their inhibitory activity against α-amylase (αA), α-glucosidase (αG), and DPP-IV, are crucial findings. However, the study would benefit from a more detailed comparison of the inhibition potencies of these extracts to existing commercial drugs. For instance, a more thorough discussion on how the inhibitory activities of the plant extracts, especially those of S. rhombifolia and S. glabra, compare with known inhibitors like acarbose and sitagliptin could provide clearer context for their therapeutic potential. Additionally, a broader discussion of the toxicity profiles of these extracts in relation to their enzyme inhibition would also help in assessing their suitability for further development as antidiabetic agents.
    • In Table 4, please check the formatting to ensure consistent units for all compounds listed under metabolite quantification.
  4. Discussion:
    • The discussion provides a comprehensive analysis, but the section comparing S. rhombifolia and S. glabra could benefit from more explicit connections between the identified metabolites and their known or potential biological activities. For instance, explaining the relevance of each compound in the context of DPP-IV inhibition would make the discussion more cohesive.
  5. Conclusion:
    • Consider adding a final statement about the broader implications of using computational approaches in bioprospecting, emphasizing how this strategy could accelerate drug discovery in other ecosystems beyond the cloud forest.
  6. Figures and Tables:
    • Ensure all figures are labeled clearly, particularly in complex diagrams like Figure 4, where docking scores for multiple compounds are presented. Annotations or arrows indicating key metabolites (like acanthoside D and cis-tiliroside) would help the reader focus on the most relevant results.

Comments on the Quality of English Language

The manuscript is generally well-written and clear, but there are some areas where sentences can be restructured for better readability. For instance, long sentences should be broken down to avoid complexity and ensure a smoother flow of ideas.

Some of the more complex sentences in the results and discussion sections could be simplified for clarity. For example, the sentences discussing the molecular docking results can be made more concise by breaking them into smaller, clearer statements.

 A few minor grammar issues (such as missing articles or incorrect prepositions) and punctuation errors were noticed. For example, some sentences are missing commas in compound sentences, which affects readability.

Reviewer 2 Report

Comments and Suggestions for Authors

The workflow used in this study presented a novel targeting strategy for identifying novel bioactive natural sources. It is interesting workflow.

Comments:

Can authors get acanthoside D and cis-tiliroside?

If these compounds actually inhibit DPP-IV enzyme, reliability of authors’ strategy is improved more.

Alpha-glucosidase inhibition:

Reference 101: Alpha-glucosidase enzyme from Saccharomyces cerevisiae (G0660, Sigma Aldrich, USA) was conventionally used. The inhibition of acarbose against this enzyme is low, meaning that authors do not evaluate the efficacy of extracts against alpha-glucosidase.

Final conclusion of this study is not change, but alpha-glucosidase enzyme from animal intestinal must be used. Refer to previous studies using rat intestinal brush borders and maltose.

Comments on the Quality of English Language

Interesting work flow is described.

Alpha-glucosidase enzyme is not suitable, but authors describe this point, it can be acceptable.

Reviewer 3 Report

Comments and Suggestions for Authors

Dear Authors, you should address the minor comments highlighted across the text.

Dear Editors, my comments related to the manuscript ‘Targeting hypoglycemic natural products from

the cloud forest plants using chemotaxonomic computer-assisted selection‘ are the following:

- The manuscript is original because it is focused on an interesting investigation which had never

been performed previously in the mentioned geographical area.

- Abstract: Fine.

- Introduction: The Authors should rephrase the last sentence, expressing it in the frame of

Introduction section not as a Conclusion like it sounds in the current form.

- Materials and Methods:

the related research topics.

- The first sub-section should regard the description of the area and years of investigation and

- Appropriate methods of analyses have been used, in my opinion.

- Results: Just a few details should be managed.

- Discussion: It has been appropriately developed, in my opinion.

- Conclusions: The content of this section correctly supports the presented research results.

- References: The citations have been exaustively reported and properly formatted.

Iasi, 20-09-2024

Sincerely,

Prof. Otilia Cristina Murariu
